# Cation-swapped homogeneous nanoparticles in perovskite oxides for high power density

Sangwook Joo[1], Ohhun Kwon[1], Kyeounghak Kim[2], Seona Kim[1], Hyunmin Kim[1], Jeeyoung Shin[3,4], Hu Young Jeong [5], Sivaprakash Sengodan[6], Jeong Woo Han [2] & Guntae Kim[1]

Exsolution has been intensively studied in the fields of energy conversion and storage as a method for the preparation of catalytically active and durable metal nanoparticles. Under typical conditions, however, only a limited number of nanoparticles can be exsolved from the host oxides. Herein, we report the preparation of catalytic nanoparticles by selective exsolution through topotactic ion exchange, where deposited Fe guest cations can be exchanged with Co host cations in $PrBaMn_{1.7}Co_{0.3}O_{5+\delta}$. Interestingly, this phenomenon spontaneously yields the host $PrBaMn_{1.7}Fe_{0.3}O_{5+\delta}$, liberating all the Co cations from the host owing to the favorable incorporation energy of Fe into the lattice of the parent host ($\Delta E_{incorporation} = -0.41$ eV) and the cation exchange energy ($\Delta E_{exchange} = -0.34$ eV). Remarkably, the increase in the number of exsolved nanoparticles leads to their improved catalytic activity as a solid oxide fuel cell electrode and in the dry reforming of methane.

[1] School of Energy and Chemical Engineering, Ulsan National Institute of Science and Technology (UNIST), Ulsan 44919, Republic of Korea. [2] Department of Chemical Engineering, Pohang University of Science and Technology (POSTECH), Pohang 37673, Republic of Korea. [3] Division of Mechanical Systems Engineering, Sookmyung Women's University, Seoul 04310, Republic of Korea. [4] Institute of Advanced Materials and Systems, Sookmyung Women's University Seoul, Seoul 04310, Republic of Korea. [5] UNIST Central Research Facilities and School of Materials Science and Engineering, UNIST, Ulsan 44919, Republic of Korea. [6] Department of Materials, Imperial College London, Exhibition Road, London SW7 2AZ, UK. These authors contributed equally: Sangwook Joo, Ohhun Kwon. Correspondence and requests for materials should be addressed to G.K. (email: gtkim@unist.ac.kr)

Exsolution has been recently explored as a method for the preparation of nanoparticles with superior catalytic activity and durability for energy conversion and storage. Specifically, exsolution refers to the formation of metal nanoparticles on the surface of a metal oxide via the release and anchoring of cations from the host lattice to the oxide surface in a reducing atmosphere, producing catalysts with enhanced lifetime compared to traditional deposition techniques (e.g., chemical vapor deposition or wet impregnation) by avoiding particle agglomeration[1,2].

Despite its benefits, the exsolution process presents two major challenges. Firstly, a significant amount of exsolved metal can remain embedded in the host bulk due to the limited diffusion rate of metal cations[3,4]. Secondly, exsolution can cause structural instability in the host material due to excessive loss of cations[5]. To overcome these challenges, several factors governing the degree of exsolution, such as the nature of the host lattice and environmental conditions[6], have been extensively investigated in simple perovskite[7,8] ($ABO_3$) or layered perovskite[9–13] ($AA'B_2O_5$). For example, A-site deficiency (A/B < 1) in perovskite oxide ($ABO_3$) has been actively researched recently in terms of cation stoichiometry/non-stoichiometry manipulation[7,14–16].

Meanwhile, topotactic ion exchange is an interesting soft chemical method that has been applied to numerous perovskite-related compounds for cation replacement[17,18]. Therefore, it could be envisaged as a solution with wide applicability for the complete exsolution of metal cations without leaving cation defects in the host lattice, thereby maintaining the overall structural features of the parent metal oxide[17].

Herein, we report the use of topotactic ion exchange to overcome the problems associated with common exsolution techniques. When a stoichiometric layered perovskite oxide ($AA'B_{2-x}C_xO_{5+\delta}$) is used, the exsolution of $y$ moles of C metal from the B site would be accompanied by the formation of the corresponding amount of B-site vacancies (Schottky-type defect) (Eq. 1). In contrast, in the topochemical ion exchange concept, such layered perovskite oxide can yield $x$ moles of exsolved C metal by the ion exchange with $x$ moles of the guest cation G (Eq. 2). Overall, the topochemical ion exchange produces the layered perovskite without B-site vacancies, thereby preserving the atomic connectivity of the B–O–B network for an efficient oxygen transport and electron conduction.

$$\underbrace{AA'B_{2-x}C_xO_{5+\delta}}_{\text{Stoichiometric layered perovskite}} \rightarrow \underbrace{AA'B_{2-x}C_{x-y}\blacksquare_yO_{5+\delta}}_{\text{Residual layered perovskite with B-site vacancy}(\blacksquare)}$$
$$+ \underbrace{yC}_{\text{Exsolved metal}},$$
(1)

$$\underbrace{AA'B_{2-x}C_xO_{5+\delta}}_{\text{Stoichiometric layered perovskite}} + \underbrace{xG}_{\text{Guest cation}}$$
$$\rightarrow \underbrace{AA'B_{2-x}G_xO_{5+\delta}}_{\text{New layered perovskite without B-site vacancy}} + \underbrace{xC}_{\text{Exsolved metal}}.$$
(2)

We selected the layered perovskite $PrBaMn_{1.7}Co_{0.3}O_{5+\delta}$ (PBMCo) as the host and Fe ($Fe^{3+}/Fe^{4+}$) as the guest cation. A previous study revealed that, in layered perovskite, the Co cation has a higher tendency to be exsolved toward the surface than Fe, mainly due to the higher co-segregation energy of Co (−0.55 eV) compared to that of Fe (−0.15 eV)[9]. Therefore, when the Fe guest cation is externally introduced into the host material, the initial host PBMCo can be converted to $PrBaMn_{1.7}Fe_{0.3}O_{5+\delta}$ (PBMFe) through topotactic cation exchange. This simple synthetic

**Table 1 Nomenclature for the compounds based on the Fe-infiltrated PBMCo system**

| Compound | Abbreviations |
|---|---|
| $PrBaMn_{1.7}Co_{0.3}O_{5+\delta}$ | PBMCo |
| $PrBaMn_{1.7}Co_{0.3}O_{5+\delta}$ + 3wt% infiltration of Fe | PBMCo-3-Fe |
| $PrBaMn_{1.7}Co_{0.3}O_{5+\delta}$ + 7wt% infiltration of Fe | PBMCo-7-Fe |
| $PrBaMn_{1.7}Co_{0.3}O_{5+\delta}$ + 12wt% infiltration of Fe | PBMCo-12-Fe |
| $PrBaMn_{1.7}Co_{0.3}O_{5+\delta}$ + 15wt% infiltration of Fe | PBMCo-15-Fe |
| $PrBaMn_{1.7}Co_{0.3}O_{5+\delta}$ + 12wt% infiltration of Co–Fe | PBMCo-12-CoFe |
| $PrBaMn_2O_{5+\delta}$ | PBM |
| $PrBaMn_2O_{5+\delta}$ + 12wt% infiltration of Fe | PBM-12-Fe |
| $PrBaMn_2O_{5+\delta}$ + 12wt% infiltration of Co | PBM-12-Co |
| $PrBaMn_{1.7}Fe_{0.3}O_{5+\delta}$ | PBMFe |
| $PrBaMn_{1.7}Fe_{0.3}O_{5+\delta}$ + 12wt% infiltration of Co–Fe | PBMFe-12-CoFe |
| $PrBa_{0.5}Sr_{0.5}Co_{1.5}Fe_{0.5}O_{5+\delta}$ | PBSCF |
| $Ce_{0.9}Gd_{0.1}O_{2-\delta}$ | GDC |
| $La_{0.4}Ce_{0.6}O_{2-\delta}$ | LDC |
| $La_{0.9}Sr_{0.1}Ga_{0.8}Mg_{0.2}O_{3-\delta}$ | LSGM |

wt%: weight percent to anode

approach not only can readily exsolve most of the cations from the bulk lattice but also can produce new compounds with multiple functionalities by exsolving nanoparticles without leaving cation defects. Moreover, we illustrate that the as-exsolved particles exhibit high catalytic activities, which are verified by solid oxide fuel cell anode test and dry reforming reaction of methane.

## Results

**System for the topotactic ion exchange/exsolution.** In this work, a layered stoichiometric perovskite, $PrBaMn_{1.7}Co_{0.3}O_{5+\delta}$, was selected as the ion exchange host for the preferential exsolution of Co to exemplify the topotactic manipulation. We selected Co as the exsolving cation since Co in the B sites has the highest co-segregation energy toward exsolution among various transition metals (Mn, Co, Ni, and Fe), whereas Fe was chosen as the guest material with the lowest co-segregation energy[9]. The deposition of guest cation was done by infiltrating a nitrate solution having different weight percentages of Fe (0, 3, 7, and 12 wt% with respect to the host material) on $Pr_{0.5}Ba_{0.5}Mn_{0.85}Co_{0.15}O_{3-\delta}$. The amount of infiltrated Fe was also calculated in a mole percentage as shown in Supplementary Table 1. After the infiltration, $Pr_{0.5}Ba_{0.5}Mn_{0.85}Co_{0.15}O_{3-\delta}$ deposited with Fe oxide was annealed in humified hydrogen at 850 °C to exsolve nanoparticles along with phase transition from simple perovskite to layered perovskite structure. Table 1 summarizes the different abbreviations of the samples.

**Ion exchange and density functional theory calculations.** In the process of Co exsolution under a reducing atmosphere, the Co cation in the host material PBMCo undergoes topotactic ion exchange with the deposited Fe due to the difference of co-segregation energy between Co and Fe. Thus, Co tends to be exsolved to the surface while Fe remains in the bulk in the $PrBaMn_{1.7}T_{0.3}O_{5+\delta}$ system (T = Mn, Ni, Co, or Fe)[9]. In a stoichiometric layered perovskite, the exsolution of transition metal cation was observed along with the phase transition under a reducing atmosphere (R1 in Fig. 1a), leaving B-site vacancies (Schottky-type defect). Under typical conditions, only a limited fraction of B-site transition metal can be exsolved. In a stoichiometric layered perovskite of $PrBaMn_{1.7}Ni_{0.3}O_{5+\delta}$ composition, only 58% of Ni can migrate to the surface, leaving many B-site vacancies[9], with the concomitant decrease in both

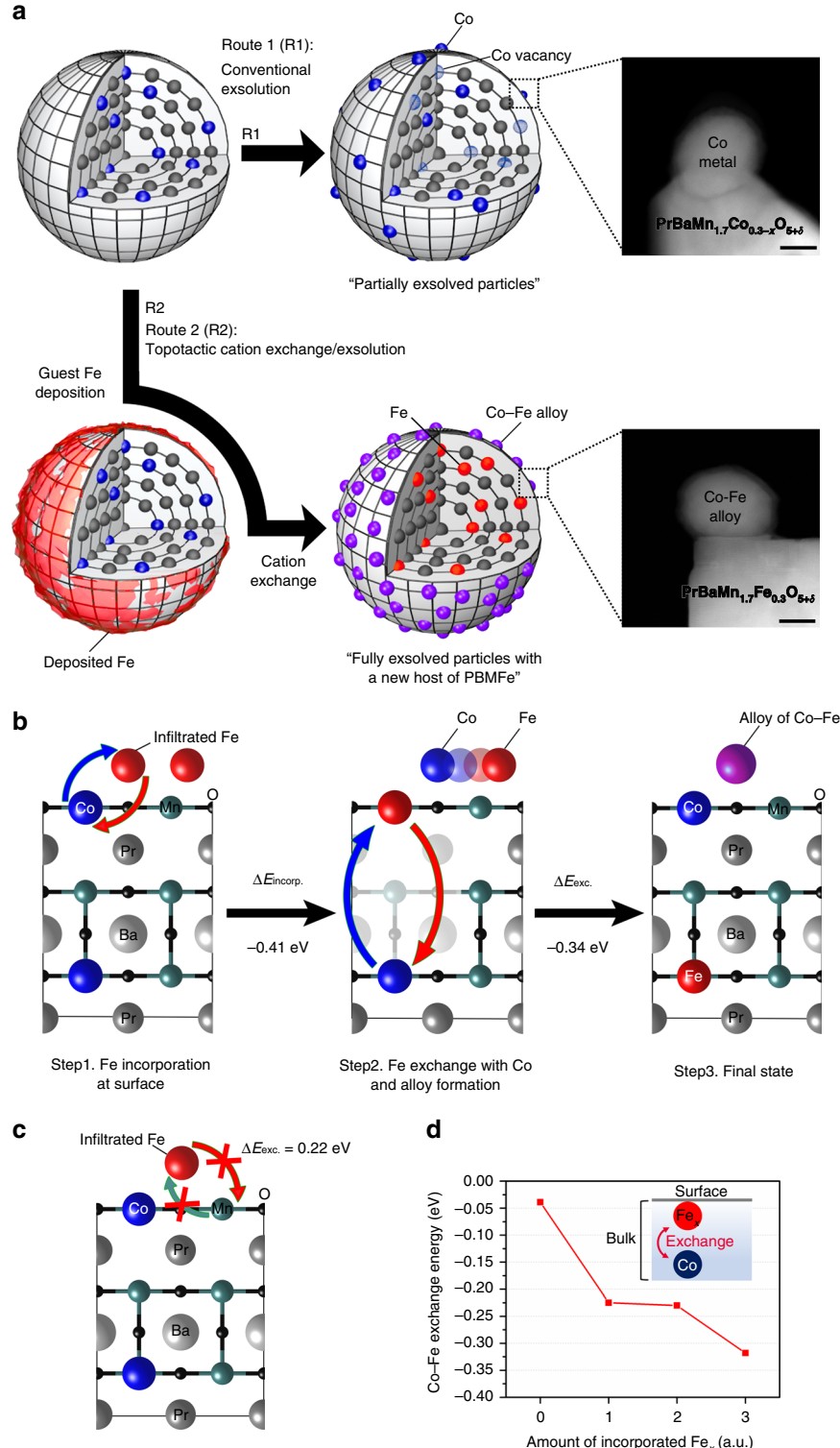

**Fig. 1** Schematic of exsolution process and density functional theory calculations. **a** Exsolution process with and without topotactic ion exchange. **b** Topotactic ion exchange energetics for the mechanism of particle exsolution via Fe infiltration on the PBMCo surface. **c** The unfavorable incorporation energy of infiltrated Fe with Mn of the top surface. **d** Calculated energetics for the Co–$Fe_x$ exchange depending on arbitrary Fe concentration

the oxygen ion conduction and electron conduction paths. On the contrary, for the topotactic ion exchange/exsolution method (R2 in Fig. 1a), the guest cation is deposited on the patent stoichiometric layered perovskite material followed by reduction. During the exsolution process, the topotactic ion exchange occurs between the lattice Co and the deposited Fe. In parallel,

all the Co cations from the B sites are exsolved without the formation of B-site vacancies. The filling of the B sites eventually leads to improved ionic and electrical conduction paths. In the topotactic ion exchange process, Fe dissolves into the underlying perovskite lattice due to its low co-segregation energy compared to that of other transition metals.

To simulate the topotactic ion exchange process between B-site cations, DFT calculation was performed. We assumed that the process occurs through two major stages, i.e. (1) incorporation of the infiltrated Fe into the lattice and (2) exchange between the incorporated Fe and the host Co, and the energy at each stage was investigated. This mechanism of cation exchange in layered perovskites can be expressed in point defect (Schottky-type defect) reactions as follows:

Exsolution without cation exchange,

$$Co_{Co}^{\times} + O_O^{\times} \leftrightarrow CoO + V_O^{\bullet\bullet} + V_{Co}'', \tag{3}$$

$$CoO \leftrightarrow Co_{(metallic\ exsolution)} + \frac{1}{2}O_2. \tag{4}$$

Exsolution by topotactic ion exchange,

$$Co_{Co}^{\times} + O_O^{\times} + FeO_{(infiltrated)} \leftrightarrow CoO_{(exsolved)} + V_O^{\bullet\bullet} + Fe_{Co}^{\times}, \tag{5}$$

$$CoO \leftrightarrow Co_{(metallic\ exsolution)} + \frac{1}{2}O_2. \tag{6}$$

where $Co_{Co}^{\times}$ denotes the Co in the Co site with net charge zero, $O_O^{\times}$ denotes oxygen in the oxygen site with net charge zero, $V_O^{\bullet\bullet}$ denotes the oxygen ion vacancy with the net charge of +2, $V_{Co}''$ denotes the cation vacancy in the Co site with the net charge of −2, $Fe_{Co}^{\times}$ denotes the incorporated Fe in the Co site with net charge zero, and FeO/CoO denotes the Fe/Co oxide, respectively.

Once Fe is deposited on the host PBMCo, Fe incorporates into the near surface of PBMCo through the exchange with the Co cations on the B sites. Since both the exsolved Co and host Mn can coexist at the near surface of PBMCo, we compared two possible exchange pathways, Fe ↔ Co and Fe ↔ Mn, on the B cation layer of the surface. Our results show that Fe ↔ Co (−0.41 eV) is thermodynamically more favored than Fe ↔ Mn (0.22 eV) (Fig. 1b, c). Thus, the incorporation of Fe occurs through its exchange with Co. After the incorporation, further exchange between the incorporated Fe and the bulk Co is thermodynamically favorable, with an exchange energy of −0.34 eV. Therefore, it can be concluded that Co exsolution is facilitated by the incorporation of Fe.

Next, the Co–Fe exchange energy was calculated as a function of the incorporated Fe concentration in an arbitrary unit (Fig. 1d). As the arbitrary concentration of the incorporated Fe increases up to the specific concentration, the Co–Fe exchange is thermodynamically favored. This also supports that the Fe incorporation into the host PBMCo possibly accelerates Co exsolution. The Gibbs energy of aggregation ($\Delta G_{aggr}$) of Co–$O_v$–Fe at the surface (surface alloy formation) is 0.01 eV, implying that the aggregation of Co and Fe requires only little energy on the surface. This result is consistent with that of the TEM investigation that will be discussed later, which evidences the formation of a Co–Fe alloy. In addition, the lower oxygen vacancy formation energy at the surface of PBMCo-12-Fe (2.52 eV) compared to that of the host PBM (2.97 eV) would promote further reduction of Co–Fe aggregation to form Co–Fe alloy nanoparticles.

**Correlation between exsolved particles and infiltration**. To provide evidence of the occurrence of topotactic ion exchange, we varied the amount of infiltrated Fe precursor and investigated the correlation between the amount of Fe deposition and the population of exsolved nanoparticles through scanning electron microscopy (SEM) and Brunauer–Emmett–Teller (BET) analysis. Figure 2a shows the schematics of the experimental process. The SEM images of PBMCo, PBMCo-3-Fe, PBMCo-7-Fe, and PBMCo-12-Fe are shown in Fig. 2b–e. The micrographs illustrate that spherical exsolved nanoparticles of 20–50 nm are evenly distributed on the surface of the parent material. Interestingly, as the amount of infiltrated Fe precursor increases from 0 to 12 wt%, more spherical particles seem to be exsolved to the surface of the layered perovskite. To provide a more quantitative correlation between the population of particles and the amount of deposited Fe, the exsolved nanoparticles in a specific area were numbered by an image analysis tool (Image J software). As seen in Fig. 2f, the results demonstrate that the amount of deposited Fe oxides promotes exsolution, particularly a significant increase up to 12 wt% of infiltrated Fe oxides. With the amount of 15 wt% infiltration, number of the exsolved nanoparticles in a specific area is not deviated from that of 12 wt% (counted as 98 particles shown in Supplementary Fig. 1a), indicating that the promotion of exsolution is saturated at the certain amount of the deposition. These trends are in good agreement with the BET analysis of the specific surface area of the material, as shown in the right axis of Fig. 2f, Supplementary Figs. 1 and 2. This can be explained by the fact that the specific surface area is affected only by the exsolved nanoparticles, not by the amount of Fe deposition. To validate this statement, we deposited Fe on $Pr_{0.5}Ba_{0.5}MnO_{3-\delta}$ and annealed it in $H_2$ to form a PBM with layered perovskite structure. The samples with 12 wt% Fe (PBM-12-Fe) and without Fe (PBM) show a specific surface area of 1.16 and 1.17 $m^2\ g^{-1}$, respectively (Supplementary Fig. 1b), and the surface morphology of PBM-12-Fe (Supplementary Fig. 1c) appears to be smooth, indicating that the contribution to the specific surface area by infiltration of 12 wt% Fe on the layered perovskite support is negligible.

**Examination of exsolved particles and parent oxide**. To investigate the crystalline structure and composition of the layered perovskite with exsolved nanoparticles, we examined the samples using transmission electron microscopy (TEM). As shown in the high-angle annular dark field (HAADF) scanning TEM image of PBMCo-12-Fe (Fig. 3a), nanoparticles having about 30 nm diameter were exsolved from the parent material. In addition, the PBMCo-12-Fe sample was subjected to energy dispersive spectroscopy (EDS) (Fig. 3b), showing that the exsolved nanoparticles consist of a Co–Fe alloy, and the parent layered perovskite contains Pr, Ba, Mn, and Fe, which is consistent with the EDS spectrum results (Fig. 3c, d). This disappearance of Co in the lattice is due to the topotactic ion exchange between the lattice Co and deposited Fe, clearly showing that Co and Fe switch their lattice positions. To gain further insights on the crystal lattice and the topotactic ion exchange, we performed atomic-scale scanning TEM analysis. The A-site ordering was observed by a small additional spot in the fast-Fourier transformed (FFT) pattern indexed to (001) of the tetragonal structure (Fig. 3e). Furthermore, atomic-scale EDS mapping was conducted in the parent oxide (orange rectangle in Fig. 3e) to investigate the A-site cation ordering and the positions of Co and Fe (Fig. 3f). It was found that the atomic positions of Pr, Ba, and Mn remained unaltered, while some Fe was observed in the position of Mn, which implies that Fe entered the B sites of PBMCo. Meanwhile, Co signals were not clearly observed in the EDS mapping, which demonstrates that most of the Co was exsolved to the surface due to the topotactic ion exchange with Fe.

Moreover, we examined XRD peaks around 22° to determine the change in lattice as exchanging cations (Supplementary Fig. 3a). The peaks around 22° corresponding to (200) are 22.79° and 22.37° for PBMCo and PBMCo-12-Fe, respectively. The peak shift to the left indicates that the lattice expansion occur due to the cation exchange of smaller Co ions ($Co^{2+}$ ($r = 0.745$ Å) or

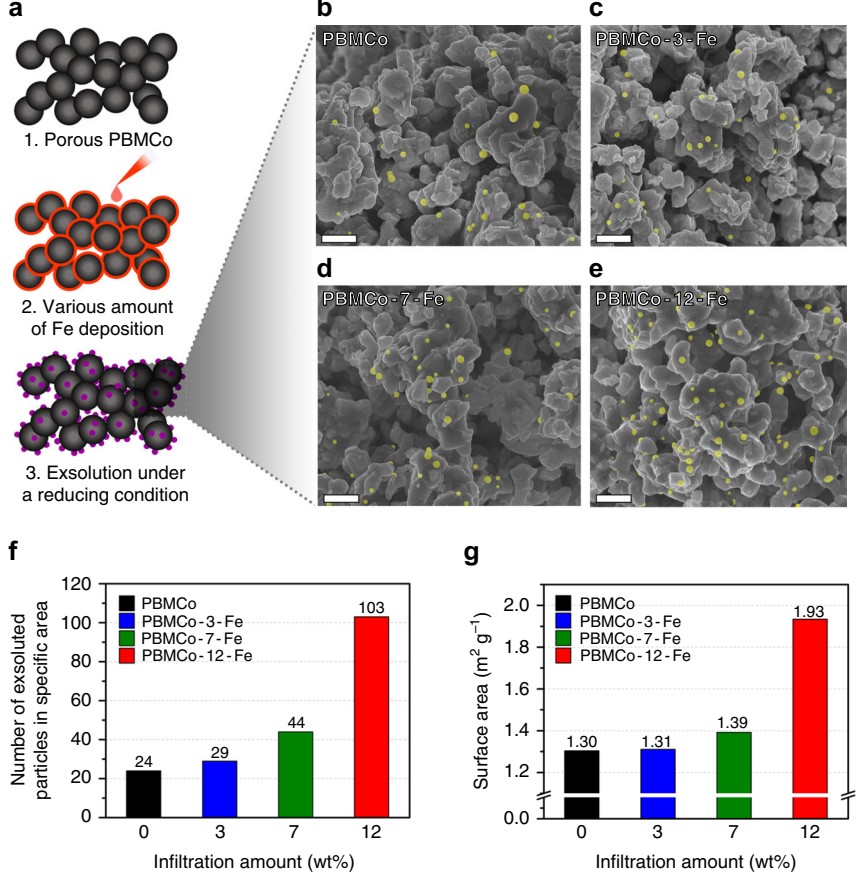

**Fig. 2** Scanning electron microscopy images and population of exsolved particles. **a** Sample preparation process for confirming the correlation between the amount of infiltrated Fe and the population of exsolved nanoparticles. **b–e** SEM images of **b** PBMCo, **c** PBMCo-3-Fe, **d** PBMCo-7-Fe, and **e** PBMCo-12-Fe (exsolved nanoparticles are highlighted in yellow); scale bars are 500 nm. **f** Number of exsolved particles in specific area counted by Image J. **g** Specific surface area calculated by the BET method

$Co^{3+}$ ($r = 0.545$ Å)) and larger Fe ions (Fe$^{2+}$ ($r = 0.780$ Å) or Fe$^{3+}$ ($r = 0.645$ Å))[19,20]. We also measured high-resolution TEM to confirm the lattice constants before and after the exchange. As shown in the HR TEM images, the lattice spaces between (001) planes of before (Supplementary Fig. 3b) and after (Supplementary Fig. 3c) exchange are identified as 0.803 and 0.815 nm by FFT pattern, respectively. Therefore, it can be concluded that the lattice constant of the layered perovskite somewhat increases after the exchange between Co and Fe.

**X-ray diffraction and X-ray photoelectron spectroscopy analysis.** The perovskite oxides were analyzed by X-ray diffraction before and after reduction. From the XRD diffraction pattern (Supplementary Fig. 4), it can be deduced that the host material samples sintered at 950 °C in air for 4 h exhibit simple perovskite structures of mixed cubic and hexagonal phases without secondary phase. The diffraction patterns of the PBMCo, PBMCo-3-Fe, PBMCo-7-Fe, and PBMCo-12-Fe samples are shown in Supplementary Fig. 5. Under a reducing atmosphere, all the samples experience phase transition from simple perovskite to layered perovskite along with the formation of exsolved nanoparticles on the surface of host materials. For PBMCo, the peak for exsolved Co metal is observed at $2\theta = 44.26°$ (JCPDS card#15-0806). As the amount of deposited Fe increases, the peak for metal is lower-angle shifted (44.26° for PBMCo and PBMCo-3-Fe and 44.17° for PBMCo-7-Fe and PBMCo-12-Fe, respectively) due to the formation of the Co–Fe alloy, which originates from the dissolution of Fe in the Co lattice[21]. The diffraction pattern of

PBMCo-12-Fe exhibits several additional peaks that are absent in those of the other perovskite oxides. This can be ascribed to the formation of PrBaMn$_{1.7}$Co$_{0.3-y}$Fe$_y$O$_{5+\delta}$ from Pr$_{0.5}$Ba$_{0.5}$Mn$_{0.85}$Co$_{0.15}$O$_{3-\delta}$ as a result of the swapping between Co and Fe cations according to Eq. (7). When the B sites of Co are fully substituted by the Fe cations, the parent material is transformed into PrBaMn$_{1.7}$Fe$_{0.3}$O$_{5+\delta}$, whose characteristic peak splitting is easily distinguishable from that of PrBaMn$_{1.7}$Co$_{0.3}$O$_{5+\delta}$[9]. These results clearly demonstrate the topotactic ion exchange between the host cation Co and the deposited Fe that leads to the selective exsolution of Co without any change in the crystal structure except the exchange of B-site cations.

$$Pr_{0.5}Ba_{0.5}Mn_{0.85}Co_{0.15}O_{3-\delta} + Fe_2O_3 \text{(deposition)}$$
$$\xrightarrow{reducing} PrBaMn_{1.7}Co_{0.3-y}Fe_yO_{5+\delta} + yCo - Fe . \quad (7)$$

(exsolution and formation of alloy)

X-ray photoelectron spectroscopy (XPS) was performed to determine the oxidation states of B-site dopants in PBMCo-3-Fe, PBMCo-7-Fe, and PBMCo-12-Fe. As shown in Supplementary Fig. 6, the binding energy peaks of Fe ions in the bulk for Fe 2p$_{3/2}$ and Fe 2p$_{1/2}$ consist of 710 and 723.7 eV corresponding to Fe$^{2+}$, 712.5 and 725.5 eV corresponding to Fe$^{3+}$, respectively. For the all samples, Fe is present as the form of mixed Fe$^{2+}$ and Fe$^{3+}$. In the case of Co, Co metal is predominant and Co$^{2+}$ and Co$^{3+}$ coexist in a similar ratio.

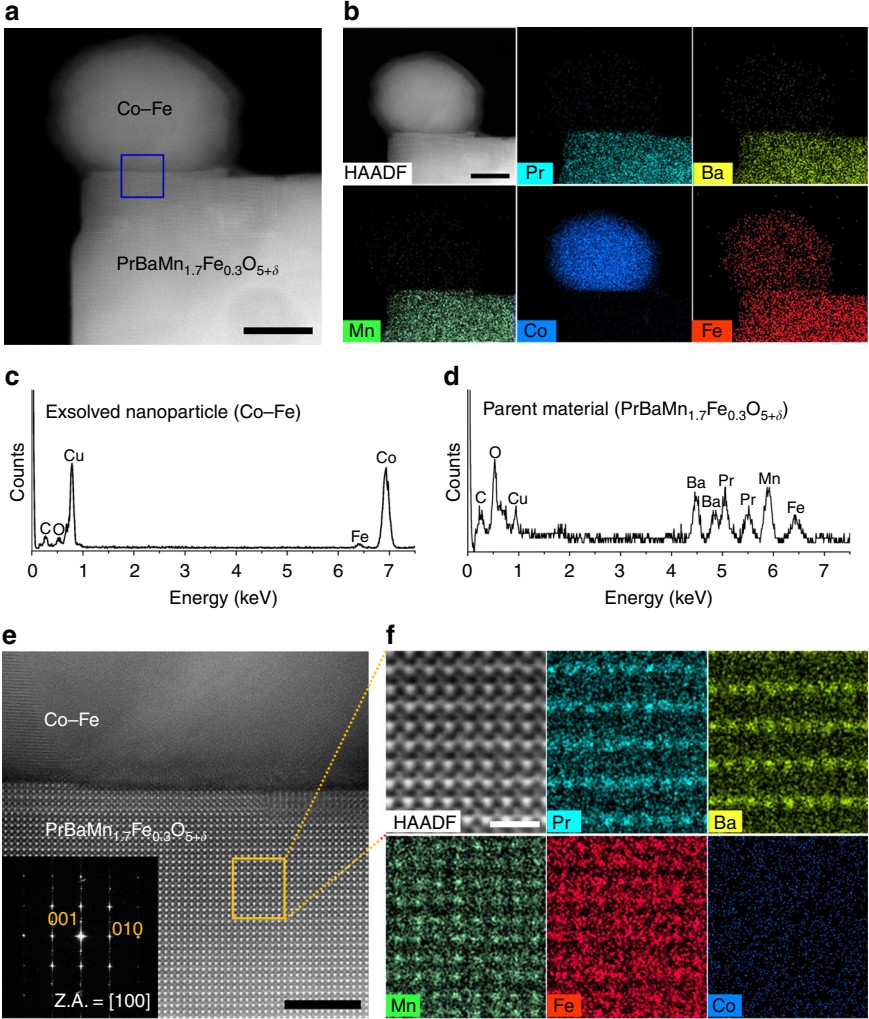

**Fig. 3** Transmission electron microscopy of exsolved particles and parent material. **a** HAADF scanning TEM image of PBMCo-12-Fe. **b** EDS elemental map of Pr, Ba, Mn, Co, and Fe; scale bar 20 nm. **c** EDS spectrum of the exsolved nanoparticles. **d** EDS spectrum of the parent material of PBMCo-12-Fe. **e** HAADF scanning TEM image of PBMCo-12-Fe (blue square in Fig. 3a) and the corresponding fast-Fourier transformed pattern with zone axis = [100]; scale bar 5 nm. **f** EDS elemental map of Pr, Ba, Mn, Fe, and Co in the parent material of PBMCo-12-Fe (yellow rectangle in Fig. 3d); scale bar 1 nm

**Catalytic activity**. To investigate the applicability of the present topotactic ion exchange/exsolution method, the electrochemical performance of fuel cells based on PBMCo-$x$-Fe as the anode was evaluated and compared with that of a PBM anode. The fuel cells with a configuration of PBMCo-$x$-Fe | LDC | LSGM | PBSCF-GDC were tested in humidified $H_2$ (with 3% $H_2O$) as the fuel and ambient air as the oxidant. The maximum power densities were 0.826, 0.853, 0.938, and 1.834 W cm$^{-2}$ for PBMCo, PBMCo-3-Fe, PBMCo-7-Fe, and PBMCo-12-Fe, respectively, at 800 °C in humidified $H_2$ (Fig. 4a). The number of exsolved particles was found to increase with the amount of Fe infiltration due to the topotactic ion exchange, which resulted in a tremendous enhancement of the electrochemical performance of the SOFC anode. In contrast, the samples without metal exsolution, i.e., the parent PBM anodes with deposited Co and Fe catalyst (Supplementary Fig. 7), showed no increment in the electrochemical performance, suggesting that the exsolved particles formed by topotactic ion exchange play a key role in the catalytic activity.

To clarify the effect of the cation exchange on the electrochemical performance, Co–Fe-infiltrated PBMFe and Co–Fe-infiltrated PBMCo were evaluated (Supplementary Fig. 8). PBMFe was used for comparative purposes to simulate the parent material after the cation exchange, since the bulk of PBMCo-12-Fe is considered to alter to PBMFe through the cation exchange. The maximum power density values of PBMFe-12-CoFe and PBMCo-12-CoFe were determined to be 0.743 and 0.962 W cm$^{-2}$, respectively, revealing that the catalytic activity of the Co–Fe alloy particles infiltrated on the parent PBMFe and PBMCo without topotactic cation exchange is not as high as that of the cation-exchanged PBMCo-12-Fe. This can be attributed to the difference in surface morphology between samples. As displayed in the HAADF scanning TEM image of the PBMFe-12-CoFe sample (Supplementary Fig. 9), the infiltrated Co–Fe alloy particles exist irregularly as coarsened particles with a size of 50–300 nm. In contrast, exsolved nanoparticles of 20–50 nm are uniformly distributed on the surface of the PBMCo-12-Fe sample (Fig. 2e). These results are in line with previous findings that present agglomeration and coarsening of catalytic nanoparticles by infiltration as well-known concerns[22].

The non-ohmic resistances for PBMCo-3-Fe, PBMCo-7-Fe, and PBMCo-12-Fe were 0.330, 0.237, and 0.071 Ω cm$^2$, respectively, at 800 °C in $H_2$ (with 3% $H_2O$) (Supplementary Fig. 10), which are consistent with the trends observed for the maximum power density. In particular, the single cell performance of

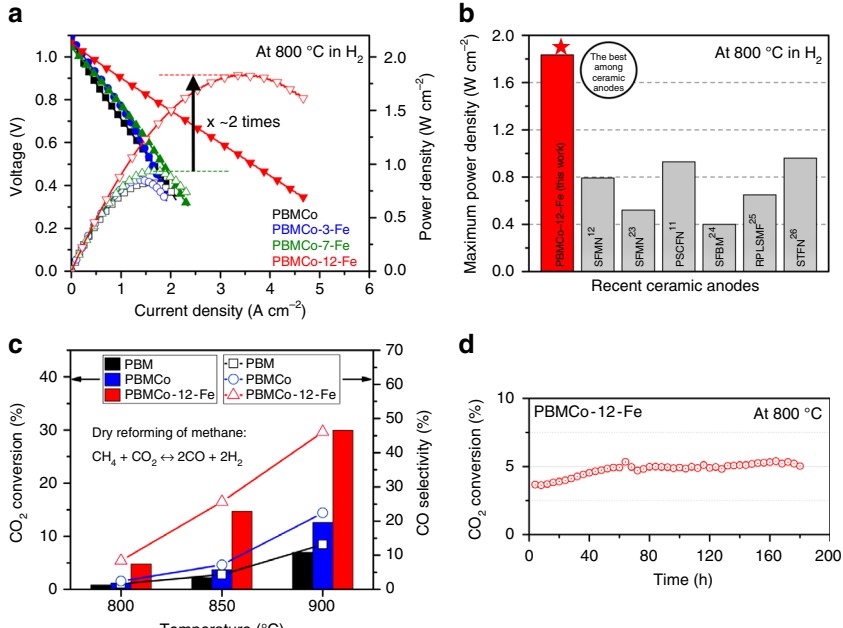

**Fig. 4** Catalytic properties. **a** *I–V* curve and the maximum power densities of the PBMCo-*x*-Fe samples. **b** Comparison of the maximum power density at 800 °C in $H_2$ from the present work and other reported studies[11,12,23–26]. **c** Conversion of $CO_2$ and selectivity of CO measured for PBM, PBMCo, and PBMCo-12-Fe in dry reforming of methane at various temperatures. **d** Time-dependence of $CO_2$ conversion for PBMCo-12-Fe in dry reforming of methane at 800 °C

PBMCo-12-Fe demonstrates superior catalytic activity among recently developed ceramic anodes using exsolution[11,12,23–26] (Fig. 4b). Additionally, to measure the stability of the particles obtained via the topotactic ion exchange/exsolution method, we compared the SEM images of the PBMCo-12-Fe sample after prolonged exposure to 3% humidified hydrogen. As can be seen in Supplementary Fig. 11, the exsolved particles maintain their morphologies without undergoing agglomeration even after exposure at 800 °C in humidified $H_2$ over 100 h.

The catalytic activity of the samples for the dry reforming of methane (DRM) was also assessed using a quartz tube reactor, since Co-based species are known to be excellent catalysts for DRM[27]. At 900 °C, the $CO_2$ conversion using the PBMCo-12-Fe sample reached 30%, which is almost two times higher than that of PBMCo and four times higher than that of PBM, as shown in Fig. 4c. The higher conversion of $CO_2$ for PBMCo-12-Fe strongly supports its excellent capability as a DRM catalyst with a long-term stability over 160 h (Fig. 4d). As shown in Supplementary Fig. 12, Co–Fe alloy has an overall metallic phase after DRM reactions and some $FeO_x$ are formed on the surface of Co–Fe alloy due to the difference in redox property of Co and Fe[1]. The reactions involving $CO_2$ oxidation and $CH_4$ reduction during DRM are given by the following steps (Eqs. (8–10)) according to a Mars-van Krevelen (MvK) mechanism[28]. That is, Co–Fe alloy particles undergo de-alloying/re-alloying process during DRM and, consequently, FeO on the surface reacts with carbon deposited on Co to form CO (Eq. (10)).

$$CH_4 \rightarrow C_{Co} + 2H_2, \tag{8}$$

$$Fe + xCO_2 \leftrightarrow FeO_x + xCO, \tag{9}$$

$$FeO_x + C_{Co} \rightarrow xCO + Co + Fe. \tag{10}$$

## Discussion

In summary, we have demonstrated the first example of a topotactic ion exchange/exsolution method that offers extensive control over the structure and properties of the obtained nanoparticles. The effectiveness of this approach emphasizes the utility of the topotactic ion exchange manipulation for the selective exsolution of catalytic nanoparticles in oxide materials. The topotactic cation exchange between Co and Fe can occur spontaneously due to the favorable incorporation energy (−0.41 eV) and exchange energy (−0.34 eV) for the deposition of the guest material Fe on the host material PBMCo, consequently resulting in the transformation of PBMCo into PBMFe, according to the results of DFT calculation. The maximum power density of an electrolyte-supported cell with a PBMCo-12-Fe anode reaches 1.834 W cm$^{-2}$ in humidified $H_2$ at 800 °C, achieving excellent electrochemical performance compared to other recently developed ceramic anodes. In addition, the catalyst activity in DRM is improved about four times and two times compared to PBM and PBMCo, respectively, at 900 °C. This approach based on topotactic cation exchange provides a powerful methodology for controlling the properties of exsolution by actively customizing the material through external cation intercalation, which goes beyond the existing methods that depend on the characteristics of the material itself.

## Methods

**Synthesis of parent materials**. $Pr_{0.5}Ba_{0.5}Mn_{0.85}Co_{0.15}O_{3-\delta}$, $Pr_{0.5}Ba_{0.5}Mn_{0.85}Fe_{0.15}O_{3-\delta}$, and $Pr_{0.5}Ba_{0.5}MnO_{3-\delta}$ were prepared by the Pechini sol–gel synthesis method. The required amounts for stoichiometry of $Pr(NO_3)_3 \cdot 6H_2O$ (Aldrich, 99.9%, metal basis), $Ba(NO_3)_2$ (Aldrich, 99+%), $Mn(NO_3)_2 \cdot 4H_2O$ (Aldrich, 98%), $Fe(NO_3)_3 \cdot 9H_2O$ (Aldrich, 98+%), and $Co(NO_3)_2 \cdot 6H_2O$ (Aldrich, 98+%) were dissolved in distilled water. After complete dissolution, proper amounts of ethylene glycol and citric acid as complexing agents were added to the solution and combustion process on heating plate is followed to make fine powders. These powders were calcined at 600 °C for 4 h to eliminate organic residue. The chemical composition of the synthesized powders and their abbreviations are given in Table 1.

**Fabrication of fuel cells**. Commercial electrolyte powders, $La_{0.9}Sr_{0.1}Ga_{0.8}Mg_{0.2}O_{3-\delta}$, (LSGM, 99.9% Kceracell) was pressed into pellet of 0.9 g and sintered at 1475 °C. After sintering, pellet was polished to about 250 µm. A buffer layer, $La_{0.4}Ce_{0.6}O_{2-\delta}$ (LDC) was prepared by ball milling stoichiometric amounts of $La_2O_3$ and $CeO_2$ (Sigma, 99.99%) in ethanol and then calcined at 1000 °C for 6 h. LDC is applied between anode and electrolyte to prevent ionic inter-diffusion. Anode powder PBMCo was mixed with an organic binder (Heraeus V006) (1:2 weight ratio) to make slurry ink. Cathode powders composed of $PrBa_{0.5}Sr_{0.5}Co_{1.5}Fe_{0.5}O_{5+\delta}$ (PBSCF)-$Ce_{0.9}Gd_{0.1}O_{2-\delta}$ (at a weight ratio of 60:40) were mixed with an organic binder (1:1.2 weight ratio) for a cathode slurry ink as described elsewhere[19,29]. These electrode inks were applied on the LSGM electrolyte pellet by screen printing method to produce a configuration of PBMCo | LDC | LSGM | PBSCF-GDC, which was followed by sintering at 950 °C in air for 4 h. The Fe precursor solution was infiltrated on PBMCo after sintering. The porous electrodes had an active area of 0.36 $cm^2$ and thickness about 20 µm. For the electrochemical tests, Ag wires were fixed to both electrodes using Ag paste as current collectors and the cell was sealed on an alumina tube using a ceramic adhesive (Cerambond 552, Aremco). The entire cell was placed inside a furnace and heated to the desired temperature. I–V polarization curves were measured using a BioLogic Potentiostat.

**Infiltration**. A deposition on sample was fulfilled by an infiltration procedure. Precursor solution for infiltration of Fe and Co–Fe were prepared in 0.7 M by dissolving an appropriate amount $Fe(NO_3)_3{\cdot}9H_2O$ (Aldrich, 98 + %), Co $(NO_3)_2{\cdot}6H_2O$ (Aldrich, 98 + %), and citric acid into distilled water. Precursor solutions were infiltrated into porous PBMCo with various weight percent to parent material (3, 7, and 12wt%) and then calcined in air at 450 °C for 4 h. This infiltration procedure was repeated to achieve the targeted weight percent.

**Exsolution characterization**. To compare the exsolution phenomenon with varying the amount of the deposited Fe on PBMCo, pre-calcined PBMCo was fired at 950 °C in air for 4 h. The sintered PBMCo was infiltrated with Fe precursor solution and reduced at 850 °C in $H_2$ atmosphere (with 3% $H_2O$) for 4 h.

The crystal structures of the samples were identified by an XRD (Bruker, D8 Advance, Cu Ka radiation, 40 kV, 40 mA). The morphologies of materials were investigated using SEM (FEI, Nova Nano 230 FE-SEM). TEM images were obtained with a FEI Titan (3) G2 60-300 with an imaging-forming Cs corrector at an accelerating voltage of 80 kV. $N_2$ adsorption and desorption isotherms measurement was carried out at −196 °C (BELSORP-Mini II, BEL Co.) to evaluate the pore structure and specific surface area. The specific surface area of the catalysts was calculated from the $N_2$ adsorption and desorption isotherms results by the BET method. XPS analyses were conducted on ESCALAB 250XI from Thermo Fisher Scientific with a monochromatic A1-Kα (ultraviolet He1, He2) X-ray source.

**Computational details**. DFT calculations were carried out using the Vienna Ab initio Simulation Package (VASP)[30,31]. Exchange-correlation energies were treated by Perdew–Burke–Ernzerhof functional based on generalized gradient approximation (GGA)[32]. An energy cutoff of 400 eV was used for plane-wave expansion. A $3 \times 3 \times 1$ Monkhorst–Pack $k$-point sampling of the Brillouin zone was used for all slab calculations[33]. Gaussian smearing was used with a width of 0.05 eV to determine partial occupancies. Geometries were relaxed using a conjugate gradient algorithm until the forces on all unconstrained atoms were less than 0.03 eV Å$^{-1}$. In order to take into account for on-site Coulomb and exchange interactions, GGA + U schemes were used with the effective $U$ values of 4.0, 3.3, and 4.0 for Mn, Co, and Fe, respectively. The eight-layered PBMO slab model was constructed with the vacuum thickness of up to 17 Å in the $z$-direction by cleaving a bulk PBMO structure[9]. The dopant position at top surface or in fifth layer represents that it is located at surface or in bulk, respectively.

In order to describe the alloy formation, we substituted two Mn atoms with Co or Fe atom in PBMO (Supplementary Fig. 13). The Gibbs free energies were also calculated for the thermodynamics of alloy and oxygen vacancy formation based on our previous calculation scheme (Supplementary Fig. 14)[9]. More calculation details are provided in Supplementary Information.

**Catalytic activity of DRM**. Catalytic activity for DRM was evaluated through gas chromatography (GC) (Agilent 7820 A GC instrument) with a thermal conductivity detector (TCD) and a packed column (Agilent carboxen 1000). The gas used for GC measurement were controlled using a mass flow controller (Atovac GMC1200) and the exact volume value of gas was calibrated through a bubble flow meter.

The 0.2 g of sample powder (950 °C sintered in air for 4 h) was prepared and packed in the middle of the quartz tube reactor using glass wool. The sample powder was in situ reduced at 900 °C for 30 min while blowing humified $H_2$ (3% $H_2O$) gas in a quartz tube reactor.

After reduction, purging for 1 h with He gas before each measurement to remove residual $H_2$, then $CO_2$, $CH_4$, and He were inserted with a ratio of 20:20:60 ml min$^{-1}$, respectively.

The dry reforming reaction is shown as below, $CO_2$ conversion and CO selectivity were calculated using the following equations[34,35].

$$CH_4 + CO_2 \leftrightarrow 2CO + 2H_2 (\Delta H^0_{298\,K} = 247\,kJ/mol)$$

$$CO_2\ conversion = \frac{[CO_2]_{consumed}}{[CO_2]_{feed}} \times 100\% = \frac{[CO]_{detect}}{[CO]_{detect}+2[CO_2]_{detect}} \times 100\%$$

$$CO\ selectivity = \frac{[CO]_{detect}}{[CO]_{detect}+[CO_2]_{detect}} \times 100\%$$

## Data availability
The data measured, simulated, and analyzed in this study are available from the corresponding author on reasonable request.

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

## Acknowledgements

This work was supported by the Korea Institute of Energy Technology Evaluation and Planning (KETEP) and the Ministry of Trade, Industry & Energy (MOTIE) of the Republic of Korea (No. 20173020032120). This work was also supported by the Mid-Career Researcher Program (NRF—2018R1A2A1A05077532) through the National Research Foundation of Korea (NRF), funded by the Ministry of Science ICT and Future Planning.

## Author contributions

S.J. and S.S. conceived the experiments and wrote the manuscript. O.K. and H.Y.J. performed TEM analysis. S.K. contributed to designing schematics. K.K. and J.W.H. performed the DFT calculation. H.K. contributed to single cell fabrication. All the authors contributed to the discussions and analysis of the results regarding the manuscript. J.S. gave constructive comments. G.K. directed the team.

## Additional information

**Competing interests:** The authors declare no competing interests.

