## [Peer Review File · Nature Communications]

Reviewers' comments:

Reviewer #1 (Remarks to the Author):

This manuscript reported the preparation of catalytic nanoparticles by selective exsolution through “topotactic ion exchange”, where deposited Fe guest cations can be exchanged with Co host cations in $\text{PrBaMn}_{1.7}\text{Co}_{0.3}\text{O}_{5+\delta}$. In addition, the catalyst activity in DRM is improved about four times and two times compared to PBM and PBMCo, respectively, at 900 °C. Solid DFT and experimental characterizations were performed to demonstrate the advantages of this method. Suggestions for improving manuscript are given below.

1. Fe was incorporated into the lattice of PBM and segregated to form the CoFe alloys. So oxidized and metallic Fe are concomitant in PBMCoFe. How about the valence states of Fe and Co?
2. Although the stable crystalline structure when Co and Fe are exchanged, the different ion radius between Fe and Co would lead to the distortion of lattice. So, how about the lattice constants before and after exchange?
3. If $\text{PrBaMn}_{1.7}\text{Fe}_{0.3}\text{O}_{5+\delta}$ was prepared previously, and then Co was used to infiltrate. Are there any differences between PBMCo +Fe and PBMFe+Co?
4. Metallic Co and Fe have different surface oxophilicity in CO₂ activation, so how about the surface CoFe alloys after DRM reactions?
5. The language needs to be further improved and some mistakes should be checked carefully.

Reviewer #2 (Remarks to the Author):

In this work, the authors reported a topotactic ion exchange as a new methodology to overcome the current problems associated with exsolution techniques. This research looks very impressive and meaningful, considering that the newly proposed method can selectively exsolve all the cations from the bulk lattice without leaving cation defects. Especially, nanoparticles prepared using topotactic exchanged exsolution exhibited high catalytic activity, which seems very interesting approach. This methodology is quite interesting and might serve as promising preparation of nanoparticles for energy conversion/storage technology.

The work presented here is substantially novel and convincing. I think that this paper would attract broad interest considering its high novelty and scientific importance, so that I recommend this

manuscript to be published in Nature Communications. Some minor points are addressed for readers' benefits.

1. The amount of infiltrated Fe is only provided as weight percent. Please provide the amount of infiltrated Fe in mole percentage for each sample. This would help readers understand the quantitative stoichiometry information.
2. The maximum amount of Fe infiltration was tested up to 12 %. What happens if the amount of infiltration exceeds 12 % (e.g., 15 %)?

Reviewer's Comments:**Reviewer #1 (Remarks to the Author):****General Comment**

This manuscript reported the preparation of catalytic nanoparticles by selective exsolution through “topotactic ion exchange”, where deposited Fe guest cations can be exchanged with Co host cations in $\text{PrBaMn}_{1.7}\text{Co}_{0.3}\text{O}_{5+\delta}$. In addition, the catalyst activity in DRM is improved about four times and two times compared to PBM and PBMCo, respectively, at 900 °C. Solid DFT and experimental characterizations were performed to demonstrate the advantages of this method. Suggestions for improving manuscript are given below.

Comment 1. Fe was incorporated into the lattice of PBM and segregated to form the CoFe alloys. So oxidized and metallic Fe are concomitant in PBMCoFe. How about the valence states of Fe and Co?

Answer 1.

We would like to appreciate your insightful comment. As per your advice, we performed X-ray photoelectron spectroscopy (XPS) to examine the oxidation states of B-site dopants in PBMCo-3-Fe, PBMCo-7-Fe, and PBMCo-12-Fe. For the all samples, Fe is present as the form of mixed Fe^{2+} and Fe^{3+} . Especially, in case of PBMCo-12-Fe, two major peaks are more developed than the other samples. In the case of Co, metal Co is predominant and Co^{2+} and Co^{3+} coexist in a similar ratio.

In section 2.5., the manuscript was revised as,

“X-ray photoelectron spectroscopy (XPS) was performed to determine the oxidation states of B-site dopants in PBMCo-3-Fe, PBMCo-7-Fe, and PBMCo-12-Fe. As shown in Fig. S6, the binding energy peaks of Fe ions in the bulk for Fe $2p_{3/2}$ and Fe $2p_{1/2}$ consist of 710 and 723.7 eV corresponding to Fe^{2+} , 712.5 and 725.5 eV corresponding to Fe^{3+} , respectively. For the all samples, Fe is present as the form of mixed Fe^{2+} and Fe^{3+} . In the case of Co, Co metal is predominant and Co^{2+} and Co^{3+} coexist in a similar ratio.”

Figure S6. X-ray photoelectron spectroscopy of (a) Fe 2p for PBMCo-12-Fe and (b) Co 2p_{3/2} for PBMCo-12-Fe.

Comment 2. Although the stable crystalline structure when Co and Fe are exchanged, the different ion radius between Fe and Co would lead to the distortion of lattice. So, how about the lattice constants before and after exchange?

Answer 2.

About the distortion of lattice as exchanging cations, we examined XRD peaks around 22 ° corresponding to (200), which were 22.79 ° and 22.37 ° for PBMCo and PBMCo-12-Fe, respectively. The peak shift to the left indicates that the lattice expansion occur due to the cation exchange of smaller Co ions (Co²⁺ (r=0.745 Å) or Co³⁺ (r=0.545 Å)) and larger Fe ions (Fe²⁺ (r = 0.780 Å) or Fe³⁺ (r=0.645 Å)) [Scientific Reports, 3, 2426 (2013)] [Acta Cryst. A32, 751 (1976)]. We also measured high-resolution TEM to confirm the lattice constants before and after the exchange. As shown in the HR TEM images, the lattice spaces between (001) planes of before (Fig. S3b) and after (Fig. S3c) exchange are identified as 0.803 and 0.815 nm by fast-Fourier transformed pattern, respectively. Therefore, it can be concluded that the lattice constant of the layered perovskite somewhat increases after the exchange between Co and Fe.

Figure S3. (a) X-ray diffraction patterns of PBMCo and PBMCo-12-Fe samples around 22 °. HR TEM image of (b) PBMCo and (c) PBMCo-12-Fe samples and the corresponding fast-Fourier transformed pattern with zone axis = [100]; scale bar 10 nm.

In section 2.4., the manuscript was revised as,

“Moreover, we examined XRD peaks around 22 ° to determine the change in lattice as exchanging cations (Fig. S3a). The peaks around 22 ° corresponding to (200) are 22.79 ° and 22.37 ° for PBMCo and PBMCo-12-Fe, respectively. The peak shift to the left indicate that the lattice expansion occur due to the cation exchange of smaller Co ions (Co^{2+} ($r=0.745 \text{ \AA}$) or Co^{3+} ($r=0.545 \text{ \AA}$)) and larger Fe ions (Fe^{2+} ($r = 0.780 \text{ \AA}$) or Fe^{3+} ($r=0.645 \text{ \AA}$)).^{19,20} We also measured high-resolution TEM to confirm the lattice constants before and after the exchange. As shown in the HR TEM images, the lattice spaces between (001) planes of before and after exchange are identified as 0.803 and 0.815 nm by fast-Fourier transformed pattern, respectively. Therefore, it can be concluded that the lattice constant of the layered perovskite somewhat increases after the exchange between Co and Fe.”

Comment 3. If $\text{PrBaMn}_{1.7}\text{Fe}_{0.3}\text{O}_{5+\delta}$ was prepared previously, and then Co was used to infiltrate. Are there any differences between PBMCo +Fe and PBMFe+Co?

Answer 3.

Thanks for your useful comment. A previous study revealed that the Co cation has a higher tendency to be exsolved toward the surface than Fe, mainly due to the higher co-segregation energy of Co (-0.55 eV) compared to that of Fe (-0.15 eV) [Nat. Comm., 8, 15967 (2017)]. Therefore, when the Fe guest cation is externally introduced into the host material, the initial host PBMCo can be converted to PBMFe through topotactic cation exchange. On the other hand, in the case of PBMFe+Co, PBMFe is not converted to PBMCo because Fe is more likely to stay in the bulk than Co. To summarize, in the case of

PBMCo+Fe, two cations change their positions, and in the case of PBMFe+Co, two cations do not change their positions.

Comment 4. Metallic Co and Fe have different surface oxophilicity in CO₂ activation, so how about the surface CoFe alloys after DRM reactions?

Answer 4.

As you mentioned, in CO₂ activation, Co remains as a metal while Fe is oxidized to FeO_x due to the difference in redox property of Co and Fe [Nat Chem 5, 916–923 (2013)]. The reactions involving CO₂ oxidation and CH₄ reduction during DRM are given by the following equations according to a Mars-van Krevelen (MvK) mechanism [ChemPhysChem 18, 3117–3134 (2017)].

That is, Co-Fe alloy particles undergo de-alloying and re-alloying process during DRM and FeO on the surface reacts with carbon deposited on Co to form CO (equation (11)). This can be confirmed by HAADF image and EDS elemental mapping of PBMCo-12-Fe after DRM. As shown in Fig. S12, Co-Fe alloy has an overall metallic phase after DRM reactions at 900 °C and some Fe oxide are formed on the surface of Co-Fe alloy due to the different surface oxophilicity of Co and Fe.

Figure S12. High-angle annular dark field (HAADF) image of PBMCo-12-Fe sample with the EDS elemental map of Pr, Ba, Mn, Co, Fe, and O after DRM test at 900 °C; scale bar

100 nm.

In section 2.6., the manuscript was revised as,

“As shown in Fig. S12, Co-Fe alloy has an overall metallic phase after DRM reactions and some FeO_x are formed on the surface of Co-Fe alloy due to the difference in redox property of Co and Fe¹. The reactions involving CO₂ oxidation and CH₄ reduction during DRM are given by the following steps (equation (9) to (11)) according to a Mars-van Krevelen (MvK) mechanism²⁹. That is, Co-Fe alloy particles undergo de-alloying/re-alloying process during DRM and consequently, FeO on the surface reacts with carbon deposited on Co to form CO (equation (11)).

Comment 5. The language needs to be further improved and some mistakes should be checked carefully.

Answer 5.

As you suggested, we have thoroughly examined the entire manuscript again and found several typos and mistakes. Also, the manuscript has been carefully reviewed by an experienced editor whose first language is English and who specializes in editing papers written by scientists whose native language is not English. The manuscript was corrected as below.

Minor fix:

- 1) In the left axis of Figure 1g, the surface area unit has been fixed as m² g⁻¹.

2) The number of equation (3) was incorrectly numbered and fixed as equation (8).

Reviewer #2 (Remarks to the Author):

General Comment

In this work, the authors reported a topotactic ion exchange as a new methodology to overcome the current problems associated with exsolution techniques. This research looks very impressive and meaningful, considering that the newly proposed method can selectively exsolve all the cations from the bulk lattice without leaving cation defects. Especially, nanoparticles prepared using topotactic exchanged exsolution exhibited high catalytic activity, which seems very interesting approach. This methodology is quite interesting and might serve as promising preparation of nanoparticles for energy conversion/storage technology.

The work presented here is substantially novel and convincing. I think that this paper would attract broad interest considering its high novelty and scientific importance, so that I recommend this manuscript to be published in Nature Communications. Some minor points are addressed for readers' benefits.

Comment 1. The amount of infiltrated Fe is only provided as weight percent. Please provide the amount of infiltrated Fe in mole percentage for each sample. This would help readers understand the quantitative stoichiometry information.

Answer 1.

Thank you for the suggestion advancing the manuscript. We calculated the amount of infiltrated Fe in mole percentage for each sample, as you suggested. The calculated values are summarized below.

Sample	Weight percent of Fe ₂ O ₃ infiltrated (%)	Mol of Fe ₂ O ₃ for the weight percentage to 1 mol of PBMCo	Mole of Fe
PBMCo-12-Fe	12	0.35	0.18
PBMCo-7-Fe	7	0.21	0.10
PBMCo-3-Fe	3	0.09	0.04
Remarks	*(Weight for 1 mol of PBMCo) = 469.28 g/mol *(Weight of 1mol Fe ₂ O ₃) = 159.69 g/mol		

Table S1. The amount of infiltrated Fe in mole percentage.

In all samples, 0.3 mol of Co is contained in 1 mol of PBMCo. Therefore, when amount of Fe deposition corresponding to 0.18, that is, more than half, the effect of topotactic exchange/exsolution can be expected.

In supplementary data, Table S1 was added. In section 2.1., the manuscript was also revised as,

“The amount of infiltrated Fe was also calculated in a mole percentage as shown in Table S1.”

Comment 2. The maximum amount of Fe infiltration was tested up to 12 %. What happens if the amount of infiltration exceeds 12 % (e.g., 15 %)?

Answer 2.

We also infiltrated PBMCo with 15 wt.%, exsolved nanoparticles are not different from 12 wt.%, which is confirmed by SEM image. For PBMCo with 15 wt.% of Fe infiltrated, number of exsolved particles in specific area counted by Image J was 98 particles in specific area, which is not varied from that of PBMCo-12-Fe.

Figure S1. (a) SEM image of PBMCo-15-Fe (Scale bar 500 nm)

In section 2.3., the manuscript was revised as,

“With the amount of 15 wt.% infiltration, number of the exsolved nanoparticles in a specific area is not deviated from that of 12 wt.% (counted as 98 particles shown in Fig. S1a), indicating that the promotion of exsolution is saturated at the certain amount of the deposition.”

In Table 1, the abbreviation for PBMCo-15-Fe was added.

REVIEWERS' COMMENTS:

Reviewer #1 (Remarks to the Author):

The authors have addressed my comments and the revised MS is recommended for acceptance.